# Large-Area Fabrication of Structurally Colored and Humidity Sensitive Composite Nanofilm via Ultrasonic Spray-Coating

**DOI:** 10.3390/polym13213768

**Published:** 2021-10-30

**Authors:** Sijun Li, Donghui Kou, Shufen Zhang, Wei Ma

**Affiliations:** State Key Laboratory of Fine Chemicals, Dalian University of Technology, No.2 Linggong Road, Gaoxinyuan District, Dalian 116023, China; lisijunok@mail.dlut.edu.cn (S.L.); koudonghui1993@mail.dlut.edu.cn (D.K.); zhangshf@dlut.edu.cn (S.Z.)

**Keywords:** composite structural color films, ultrasonic spray-coating, large-area, hydrophilic property, humidity sensitive

## Abstract

Intelligent structural colors have received extensive attention in recent years due to their diverse applications. However, the large-area, uniform, and cost-effective fabrication of ultra-thin structural color films is still challenging. Here, for the first time, we design and employ an ultrasonic spray-coating technique with non-toxic, green nano-silica and polyvinylpyrrolidone as raw materials, to prepare structural color films on silicon wafers. Due to the addition of polyvinylpyrrolidone, the coffee-ring effect during droplet drying is suppressed and uniform composite films are formed. We further perform a detailed study of the influence of various processing parameters including silica/polyvinylpyrrolidone concentration, substrate temperature, nozzle-to-substrate distance, and number of spray-passes on film roughness and thickness. By increasing the number of spray-passes from 10 to 30, the film thickness from 120 to 340 nm is modulated, resulting in different colors, and large-area and uniform colors on commercial round silicon wafers with 15 cm diameter are achieved. The silica/polyvinylpyrrolidone composite films show strong hydrophilicity and are sensitive to humidity changes, leading to quickly tunable and reversible structural colors. Quartz crystal microbalance with dissipation demonstrates water vapor adsorption and condensation on the nanofilm when increasing environmental humidity. Thereby, ultrasonic spray-coating as a novel film fabrication technique provides a feasible scheme for large-area preparation of intelligent structural colors.

## 1. Introduction

Replacing chemical colors with artificial structural colors has attracted more and more attention due to contamination from chemical dyes and pigments, and easy fading of chemical colors [1,2]. Structure color, which arises from the physical interaction of visible light with micro- and nano-fabricated ordered structures, has a lot of advantages, such as having good light stability, and being pollution-free and fadeless [3,4]. However, the superior advantage that chemical colors can be manufactured on an industrial scale makes it difficult to find substitutes for them [5,6]. There are still enormous challenges to replace chemical colors with artificial structural ones [7].

Attention has been paid to thin-film interference due to its use of a simple structure to achieve color control through the adjustment of layer thickness and refractive index [8,9]. The phenomenon occurs when the incident light wave is reflected off each boundary of the membrane and the two reflected waves interfere with each other to form a new wave. When the membrane is uniform and its thickness is comparable with the wavelength of visible light, structural colors are produced [10].

In nature, some animals are colored by the principle of film interference, showing a variety of attractive colors. Some squids can produce iridescent colors, which are caused by the light reflection of platelet stacks and can be adjusted with the thickness change of the platelets [11]. Moreover, living organisms can change their colors for communication and camouflage by tuning the thickness or volume of their protein soft layer or cell [12]. For example, the longhorn beetle changes the color of its elytron from green to red when the environmental humidity increases due to water vapor adsorption [13]. Inspired by the intelligent colors in nature, artificial colors have been constructed for diverse applications [14,15]. Previously, the reported ultra-thin films have mainly been assembled via spin- or dip-coating, or fabricated utilizing atomic layer deposition (ALD) or sputter deposition techniques [16,17,18]. However, these methods have limitations in scale-up and economical preparation. Thus, large-scale and low-cost fabrication of structurally colored smart nanofilm is highly desirable and remains challenging.

Spray-coating is an alternative and promising way to realize the large-scale objective because of its advantages of economical nature, convenience, practicability, and ease of scaling up to large production volumes [19]. However, traditional spray-coating with air pressure for atomization cannot achieve accurate control of layer thickness in the nanoscale due to larger droplet sizes [20]. In recent years, another novel technique (ultrasonic spray-coating) has developed quickly. With this technique, high-frequency ultrasonic vibration produces a fine mist of solution with droplets of 10–30 μm in size, which is beneficial for precise control of film thickness [21,22,23,24]. Although this method shows many advantages, achieving a uniform thin layer with structural colors is still challenging and has not previously been reported. During the spray process, the formed droplets are deposited on a substrate where they coalesce to form a coating [25]. However, when a droplet containing nonvolatile solutes dries on a solid surface, it leaves a dense, ring-like deposit along the perimeter, forming a “coffee ring” on the surface, which greatly affects the uniformity and color of the film [26,27].

A drop of evaporating water is a difficult-to-control, complex, non-equilibrium system [28]. Except for capillary flow, the evaporating droplet features a gas–liquid interface shaped like a spherical cap and Marangoni flows caused by surface tension gradients between the top of the droplet and the contact line [29]. Attempts to ameliorate or reverse the coffee-ring effect have focused on manipulating capillary flows and Marangoni flows [30]. Liying Cui et al. reported that adding water-soluble polymers to microsphere dispersion could effectively suppress the coffee-ring effect [31]. The polymer additives cause contact line motion due to their viscosity and the Marangoni effect, which results in a reduced amount of solutes depositing on the edge of the droplet. In addition, depositions from water drops containing nanoparticles and different concentrations of sodium dodecyl sulfate have been investigated [32]. As the droplets evaporate, the concentration of the surfactants increases locally, generating strong gradients in the surface tension to generate Marangoni flows, which can prevent most solutes from reaching the contact line, leading to a more uniform deposition of nanofilm.

Herein, for the first time, we use the ultrasonic spray-coating technique, with nano-silica and polyvinylpyrrolidone (PVP) to fabricate large-area tunable structure colors on silicon wafers. SiO_2_ is a widely used inorganic film-forming material, and it shows the properties of non-toxicity, good stability, and hydrophilicity [33,34]. PVP is a kind of biocompatible nonionic polymer; it can assist in dispersing nanoparticles and adjust the viscosity of the system [35]. The incorporation of PVP in SiO_2_ dispersion will benefit the formation of the homogeneous film. The processing scheme for nanofilm preparation and schematic of PVP adsorption on the surface of silica particles are presented in Figure 1a. This shows that under the action of a carrier gas, small droplets are formed through the ultrasonic nozzle, which is sprayed onto the substrate; through repeated cycles, a film with a certain thickness is achieved. The nozzle generates a fine mist of droplets by high-frequency vibration, as shown in Figure 1b. The color is generated by the interference of the two reflected light waves from the air–nanofilm, and film–silicon substrate interfaces (Figure 1c). In this study, droplet drying processes were monitored under a microscope, and a comparison was made between SiO_2_ and SiO_2_/PVP droplets. Processing parameters were optimized to achieve low surface roughness. Through adjusting the number of spray passes, tunable structural color films with different sizes were achieved. In addition, it was found that the as-prepared films showed strong hydrophilicity and were very sensitive to humidity changes, demonstrating their excellent visual sensing functionality.

## 2. Materials and Methods

### 2.1. Materials

The nano-silica (SiO_2_) (30.0wt%, Aladdin, Shanghai, China), Polyvinylpyrrolidone (PVP) average molecular weight 5.5 × 10^4^ (Shanghai Qiangshun Chemical Reagent, Shanghai, China). Round silicon wafers (100) with double-sided polishing and a diameter of 15 cm (Lijing Silicon Material, Hangzhou, China), were cut into small squares with a size of 2.0 cm × 2.0 cm, and then treated with plasma to be hydrophilic.

### 2.2. Preparation of Spray Solution

SiO_2_ sol was first obtained by dispersing SiO_2_ (30 wt%) into a certain amount of deionized water and then uniformly dispersing via ultrasonic treatment. In the dispersion, the mass concentrations of SiO_2_ and PVP were 0.05 wt% and 0.01 wt%, respectively. We simplified the solution as SiO_2_/PVP (0.05/0.01).

### 2.3. Process Parameter Optimization of Spraying Equipment

Spray coating was performed on a UC340 ultrasonic spraying machine (Dongfang Jinrong, Beijing, China) equipped with a D12 ultrasonic nozzle. The main components are illustrated in Figure 1a. The spraying solution was placed in a syringe pump connected by tubing to the atomizing nozzle. The pump was programmed to a constant liquid infusion rate of 0.1 mL/min. The tip of the ultrasonic atomizer nozzle was actuated at a frequency of 120 kHz with a generator power of 1.5 W. The movement of the nozzle was controlled by an x−y−z stage, and the nozzle path as shown in Figure 1. The distance between the two nozzle paths was kept at 2 mm, which is smaller than the width of the spray (~4 mm). The substrate to be coated was placed on the hot plate, and the temperature was varied according to the experimental design. Three values of each variable were selected with the concentration of the spraying solution (SiO_2_/PVP) (0.05/0.01, 0.1/0.02, 0.15/0.03, and 0.2/0.04), the substrate temperature (40, 50, and 60 °C), and the distance between the nozzle and substrate (55, 60, 65, and 70 mm) for the spraying experiment on the silicon wafer with a size of 2.0 cm × 2.0 cm. Spraying experiments with optimized parameters were carried out on a round silicon wafer with 15 cm in diameter.

### 2.4. Humidity Response

Humidity response tests were performed in a sealed container with a digital temperature-humidity recorder. Different amounts of water was added to the sealed chamber of the sample to be tested and converted into water vapor with different relative humidity. The reflectance spectra were recorded after the maximum reflection wavelength was kept constant and the photographs of the samples were captured by a camera (Nikon). All tests were performed at room temperature.

### 2.5. Characterization

Particle hydrodynamic size, distribution, and zeta potential of SiO_2_ nanoparticles were measured with a particle size analyzer (Zetasizer nano series Nano-ZS100, Malvern, London, UK). The thickness and roughness of the film were measured by an NV5000 profilometer (Zygo Company, Middletown, Connecticut, USA). The cross-section and surface analysis of the films were measured by scanning electron microscopy (SEM, SU8220, Hitachi, Tokyo, Japan). An optical microscope (Autor optics MIT series metallographic microscope) was used to monitor the process of droplet drying. The reflectance spectra of the structural color film at an incident angle of 5° were measured by a spectrophotometer (U-4100, Hitachi, Tokyo, Japan) and vertically using a fiber-optic spectrometer (PG 2000, Ideaoptics, Shanghai, China). The refractive indexes of the SiO_2_ layer were measured by an Ellipsometer type Ellip-ER-III under a certain wavelength (632.8 nm). The structured color film was taken with a camera (Nikon) at 5° for digital photos. The water contact angles were measured by using a contact-angle system (JC2000D1, Powereach, Shanghai, China). Surface morphology was measured by using an atomic force microscope (Dimension Icon, Bruker, Madison, WI, USA). Mass and viscoelastic change of the SiO_2_ film were monitored by a quartz crystal microbalance with dissipation (QCM-D, E1 model, Biolin Scientific, Gothenburg, Sweden).

## 3. Results

### 3.1. Influence of PVP on Assembly of the Nanofilm

The properties of SiO_2_ dispersions without and with PVP were first compared. Figure 2a shows that the average particle size and PDI of the SiO_2_ are 39.09 nm and 0.197, respectively, and its zeta potential is −32.1 mV. SiO_2_ dispersion in Figure 2a is transparent and presents good stability at room temperature. In addition, the surface tension of SiO_2_ dispersion is 71.7 mN/m (Appendix A) and its contact angle on a silicon wafer is 35° (Figure 2c). As displayed in Figure 2b, the average particle size and PDI of SiO_2_/PVP are 40.26 nm and 0.262, respectively, and its zeta potential is −35.4 mV. The surface tension of SiO_2_/PVP dispersion is 68.8 mN/m (Appendix A) and its contact angle on a silicon wafer is 28° (Figure 2d). This shows that the addition of PVP does not affect much the particle size and surface charge, and SiO_2_/PVP is still stable and transparent. Besides, the viscosities of the dispersions at different temperatures were measured as shown in Appendix A. This presents the addition of PVP; the viscosity of the solution increases at all temperatures, which is attributed to the larger viscosity of the polymeric PVP.

The coffee-ring phenomenon produced after droplet drying has an essential influence on the uniformity of the nanofilm, so the droplet drying process was studied and monitored. During the drying process, there are three main flow patterns of nanoparticles in evaporated droplets [36]. The first flow pattern is the transport of particles normally toward the substrate, occurring in the case of gravity, as shown in Figure 3a. The second relevant flow pattern is the radial flow caused by the maximum evaporation rate at the pinned wetting line as shown in Figure 3b. The third flow pattern is a Marangoni flow presented in Figure 3c. We used SiO_2_ (0.05) and SiO_2_/PVP (0.05/0.01) as spraying solutions, and the sprayed droplets were placed under an optical microscope for observing the drying process. The drying process of SiO_2_ droplets is shown in Figure 3d and Appendix A. The droplets spread out in circles of about 60 μm in diameter on the silicon wafer and took 3.6 s to dry. After drying, the surface left a ring of sediment, producing a distinct coffee-ring phenomenon.

For SiO_2_/PVP dispersion, the diffusion area of droplets is also about 60 μm and the drying time is 3.5 s, as shown in Figure 3e and Appendix A. After drying, the coffee ring on the edge turned out to be unobvious. As can be seen from the dynamic change of the optical image in Figure 3e and the video of the drying process, the fluidity of the solution is enhanced, and the liquid at the edge of the droplet flows gradually to the center, reducing the droplet aggregation at the edge. As a kind of polymer surfactant, and addition of PVP can decrease the surface tension of the droplets (see Appendix A) and form a difference in surface tension between the droplet edge and the center during the drying process. This promotes Marangoni flow, which is the dispersion at the edge of the droplet flows to the center. In addition, the increased viscosity due to the addition of PVP also resists radial flow of the solute from the center to the droplet edge. Therefore, the promoted Marangoni flow and increased viscosity weaken the coffee-ring effect and are beneficial for forming a uniform film.

We used SiO_2_ and SiO_2_/PVP as spraying solutions to conduct spraying experiments with spray passes of 10. By spraying SiO_2_, we can see a rough surface from its SEM image in Figure 4a. The profilometer scan shows that the surface morphology is rough as presented in Figure 4c. Ra (*Ra =*
$\overset{N}{\underset{n=1}{\sum }}\left|Z_{n}-\bar{Z}\right|/N$, *Z_n_* is the individual height value of the measuring point, and $\bar{Z}$ is the average value of all height points) is 38.1 nm [37]. As shown in Figure 4b, the SEM image of SiO_2_/PVP film with 10 spray passes shows a uniform surface and the coffee-ring phenomenon is suppressed. The profilometer scan also demonstrates that the SiO_2_/PVP surface is homogenous and the obtained Ra is 10.5 nm, as shown in Figure 4d. Taking the above spraying results into consideration, the SiO_2_ and PVP composite was further used as the spraying solution.

### 3.2. Process Parameter Optimization of Ultrasonic Spraying Coating

The uniformity of a film by ultrasonic spray-coating is also influenced by process parameters that need to be regulated to assemble a uniform film [38]. In this work, we performed a detailed study of the influence of the concentration of the solution, the temperature of the substrate (T), the nozzle–substrate distance (H), and the number of spray passes (N) on Ra and thickness of the spray coating.

Figure 5a and Appendix A show the results of Ra and thickness when SiO_2_/PVP concentrations are 0.05/0.01, 0.10/0.02, 0.15/0.03, and 0.20/0.04 wt%/wt%, respectively; T is 50 °C and H is 60 mm. It can be seen that at a close film thickness of about 340 nm, the solution concentration has a distinct influence on the layer roughness. When the SiO_2_/PVP concentration is 0.05/0.01 wt%/wt%, Ra is 24.1 nm; when we further increase the concentration to 0.10/0.02, 0.15/0.03, and 0.20/0.04 wt%/wt%, Ra increases to 33.4, 45.8, and 48.7 nm, respectively, indicating that the film becomes rougher. The reason for this result is that the higher the solution concentration, the higher the solute content in the small droplet, and the stronger the coffee-ring effect that will be produced in the drying process.

In Figure 5b and Appendix A, keeping the concentration of SiO_2_/PVP at 0.05/0.01 wt%/wt%, H = 60 mm and the number of spray passes (N) of 10, the substrate temperature increases from 40 °C to 50 °C, then to 60 °C. This shows that when the temperature is 40 °C, the average layer thickness is 121.2 nm and Ra is 25.8 nm; when the temperature is 50 °C, the thickness is 129.7 nm and roughness decreases to 10.5 nm; and when the temperature increases to 60 °C, thickness and Ra change to 123.6 nm and 30.9 nm, respectively. It can be observed that with the increase in the substrate temperature, the thickness of the film does not change significantly, while the roughness reaches a minimum value at 50 °C. In the early stages of evaporation, some nanoparticles move outward but do not deposit at the edge. These nanoparticles flow back to the droplet center at the edge of the droplet and form a circular collection of nanoparticles near the gas–liquid interface. However, with the increase in temperature to 50 °C, the Marangoni flow was strengthened, and the flow of the solution at the edge of the droplet was strengthened to the center, so a more uniform deposition was formed after drying. However, when the temperature further rised to 60 °C, it was found Ra was increased, which was mainly due to that higher temperature accelerated the volatilization of water, leading to less even distribution of the solute and increased surface roughness.

Figure 5c and Appendix A give the effect of nozzle-to-substrate distance on the changes in surface roughness. SiO_2_/PVP (0.05/0.01) was also used, the substrate temperature was kept at 50 °C, and the investigated distances between the nozzle and the substrate were 70, 65, 60, and 55 mm. The results are that Ra is 20.2, 19.1, 10.5, and 13.4 nm, and the film thickness is 123.1, 124.6, 129.7, and 121.5 nm, respectively. The minimum Ra value of 10.46 nm was achieved at the height of 60 mm. With the decrease in the distance between the spray nozzle and the substrate, the force of the droplet impacting the substrate becomes larger, and the droplets are well spread on the substrate, weakening the coffee-ring effect. However, when the nozzle–substrate distance drops to 55 mm, the impacting force further increases, and the droplet slides and springs on the substrate during spraying, which affects the uniformity of the surface.

Figure 5d and Appendix A show Ra and thickness scanned by profilometer at the number of spray passes of 10, 20, and 30. The other conditions used are SiO_2_/PVP (0.05/0.01), T = 50 °C, and H = 60 mm. The surface roughness of N 10, 20, 30 is 10.5, 14.5, and 24.1 nm, respectively, and the thickness is 129.7, 224.9, and 337.2 nm, respectively. Their Ra increases with the increase in the number of spray passes. There is a linear relationship between the thickness and the number of spray passes. Therefore, we can control the film thickness by controlling the number of spray passes. Based on the above results, the solution concentration of SiO_2_/PVP (0.05/0.01), substrate temperature of 50 °C, and nozzle-to-substrate distance of 60 mm were employed in the following spray coating studies.

Figure 6a shows an even cross-sectional SEM image of a SiO_2_/PVP film when the spray pass is 30 and the film thickness is measured to be 337 nm. The surface SEM image of the film presented certain small pits. Just as shown in Figure 6b, spraying defects become apparent as the number of spray passes increases. Then, a high magnification SEM image of SiO_2_ shows particle morphology as depicted in Figure 6c and PVP fills the spaces between the particles. Energy dispersive X-ray (EDX) mappings and the spectrum of C, O, and Si elements derived from SEM images (Figure 6d) demonstrate that all these elements are distributed throughout the surface uniformly.

### 3.3. Optical Properties of Structure Color Films

During the assembly process of the film, we controlled the thickness by changing the number of spray passes, preparing different structural color films. In addition, it was measured that the refractive index of the SiO_2_/PVP layer was 1.38. The theoretical refractive indices of air and silicon are 1 and 3.8, respectively [39]. In this case, the condition for constructive interference is *2ndcosθ = mλ*, where *d* is the thickness of the SiO_2_/PVP film, *θ* is the angle of the incident wave, *λ* is the wavelength of interference light, *n* is the refractive index, and *m* is an integer [40]. Figure 7a shows that when N is 20, 22, 24, 26, 28, and 30, the thickness of the film is 225.6, 248.3, 270.1, 295.4, 313.4, and 337.2 nm, respectively, as shown in Appendix A, and the reflection wavelength is 424.7, 471.5, 525.4, 549.4, and 602.3 nm, respectively. Their digital photographs present purple, blue, green, yellow-green, brown, purplish red, respectively. The reflectance spectrum changes of a SiO_2_/PVP film (N = 26) by varying the incident angles are shown in Figure 7b. When the observing angles are 5°, 25°, 35°, 45°, and 75°, their digital photographs present green, blue, light blue, light red, and red, respectively. This shows that as the incident angle continues to increase, the position of the reflection peak gradually blue-shifts. These are both in good accordance with the thin-film interference law. With the previously optimized conditions, we carried out the spraying experiment on round silicon wafers with diameters of 15 cm, and prepared three pieces of SiO_2_ film with green, red, and blue colors, as displayed in Figure 7c.

### 3.4. Hydrophilic Property and Humidity Responsiveness of the Nanofilms

In addition to coloration and large-area preparation, we also found that the films had strong hydrophilicity. The dynamic change in water contact angle and the surface morphology of the films with different spray passes were studied and the results are shown in Figure 8. As shown in Figure 8a, after 1.6 s, the water contact angle of the 10-pass film decreased from the initial 25° to 11.5° and reached a stable level. The AFM image in Figure 8d shows that the film has a uniform surface; Figure 8b presents that the water contact angle of the film with N = 20 decreases from 23° to 9° and its surface morphology becomes uneven and rough as displayed in Figure 8e. Figure 8c shows that the water contact angle of the film with N of 30 is only 8° and the AFM image presents an even rougher surface and small holes as displayed in Figure 8f. The above results demonstrate that as the thickness of the film increases, both surface roughness and hydrophilicity increase. These unique surface wetting properties are attributed to the water adsorption properties of mesoporous structures and the hydrophilic SiO_2_/PVP composite, which benefit the diffusion and penetration of water within the film.

We found that the film had a very sensitive color change response to environmental humidity as shown in Figure 9. The sensing properties for various relative humidity were monitored by comparing the structure colors and recording the reflectance spectra. Figure 9a shows that with the relative humidity (RH) increased from 33% to 45%, 75%, 86%, and 97%, the color of the film quickly turned from violet to green, orange, light pink, pink and blue, respectively, realizing humidity response discoloration. In Figure 9a, the reflected wavelength varied from 487.6 to 541.1, 581.6, 639.6, and 678.2 nm. An obvious bathochromic shift was observed when the film was stimulated by water molecules. The mechanism for color change by varying humidity is based on the change in refractive index of the film, which results from the adsorption and penetration of water vapor within the thin coating. Based on the coloration mechanism of thin-film interference, as Equation *2ndcosθ = mλ* shows, the reflection wavelength *λ* of the film is proportional to the refractive index *n*. When water molecules penetrate into the film, its refractive index increases as the air within the film is replaced by the water vapor, resulting in a color change and red-shift of the reflection wavelength. Due to the color change of the intelligent film upon exposure to different RH, we realize naked-eye recognition of humidity change.

In addition, the response and recovery time of the humidity sensor were tested based on human blowing. The process is shown in Appendix A. As can be seen from the video, upon exposure to the blowing, the color of the sample immediately changed, from blue-green to light yellow, and then to pink red, which took 0.9 s. After stopping blowing, the color quickly changed back to blue-green; the recovery time was about 0.6 s. Therefore, the total time was only 1.5 s, demonstrating a very sensitive humidity response. The color variation mainly resulted from the change in reflective index of the film after adsorption and desorption of water, and the very fast response and recovery are attributed to the mesoporous structure of the hydrophilic nanofilm, which facilitates rapid absorption and release of the water molecules. Appendix A shows ten cyclic response tests of the structural color film at RH of 33% and 97%. The positions of the reflection peaks change reversibly from about 480 nm to 680 nm, indicating good cyclic stability of the film for humidity response.

To better understand the sensing and adsorbing dynamics, quartz crystal microbalance with dissipation (QCM-D) monitoring was implemented for in situ measurements of mass changes in the range of nanograms as a frequency shift (Δf) and of viscoelastic changes as a dissipation shift (ΔD). For example, Δf is expected to decrease when SiO_2_ films attached to the QCM-D sensor adsorb water vapor, and the value of ΔD will rise if a viscoelastic property of the attaching layer increases, which can directly demonstrate the adsorption and condensation of water vapor on the surface of the film. Variations in Δf and ΔD of the SiO_2_ film assembled on a gold-coated chip when exposed to different relative humidity are displayed in Figure 9c.

As visualized in Figure 9c, we used saturated LiCl, MgCl_2_, NaCl, and K_2_SO_4_ solution to create 11.3%, 32.7%, 75.3%, and 97.3% humidity environments at 25 °C, respectively [41]. At the initial stage (Regime I), LiCl solution was injected to produce an 11.3% water vapor environment; Δf and ΔD reached balances rapidly, which were set as the reference baselines (Δf = 0; ΔD = 0). Then MgCl_2_ saturated solution was injected into the chamber to produce a 32.7% water vapor environment; the Δf decreased obviously and immediately owing to the increased mass of the film that stemmed from the strong water absorption capacity of the SiO_2_ films with an uneven surface in the 3D AFM images in Figure 9b. As less water vapor enters the pores on the surface of the SiO_2_ film, ΔD of the film does not change. The time to reach adsorption equilibrium is 29 s. Subsequently, NaCl saturated solution was introduced into the testing system to produce a 75.3% water vapor environment and the data curves of Δf further decreased and ΔD has no significant change in values. We can see that the water vapor adsorption mass of the film decreases and reaches equilibrium in a shorter time. Next, K_2_SO_4_ saturated solution was introduced into the testing system to produce a 97.3% water vapor environment. The Δf experienced a two-step gradient descent. It first decreased to −527 Hz within 80 s, then quickly decreased to −1032 Hz; for ΔD, in the first 80 s, when RH changed to 97.3%, it still kept unchanged, and then it directly increased to 83 × 10^−6^. The distinct changes in both Δf and ΔD are presumably owing to water vapor condensation in the film. The vapor was first adsorbed on the surface, then entered the holes; when the RH was high enough, water vapor began to condense, resulting in a viscoelasticity increase. When the saturated LiCl solution was again passed through to create a humidity of 11.3% (Regime II), the film was restored to the initial stage, which proves that the adsorption of water vapor on the film is fully reversible.

For comparison, the humidity response of spin coating films with the same thickness was prepared and a QCM-D humidity test was conducted. As shown in Appendix A, the adsorption capacity of water vapor on the spin-coated SiO_2_/PVP film is much weaker than that of the sprayed one when the RH changes from 11.3% to 97.3%, and the Δf change is only 120 Hz, which is much less than the 1051 Hz change of the sprayed film. The reason for this is that the surface of the SiO_2_ film prepared by spin coating is dense, which is not conducive to the adsorption of water vapor.

## 4. Conclusions

In summary, structural colors show potential applications in colorimetric sensing, display, and coating; however, due to the difficulty in large-area manufacturing of the smart colors, their application is limited. In this study, a recently developed technique (ultrasonic spraying-coating) was employed and large-area ultra-thin structural color films were successfully constructed. For the first time, the drying process of ultra-sonic spray droplets was tracked by an optical microscope, and it was confirmed that PVP as an auxiliary can inhibit coffee rings and reduce roughness. Meanwhile, the experiments confirm the optimal process parameters: the concentration of SiO_2_/PVP is 0.05 wt%/0.01 wt%, the optimal substrate temperature is 50 °C, and the optimum nozzle distance is 60 mm. By controlling the number of spray passes, we prepared uniform structural color films with 15 cm diameters. The nanofilms are intelligent and show quick color changes in response to humidity variations, which is attributed to their mesoporous and hydrophilic structures. QCM-D results prove that the composite film has a stronger ability to adsorb water vapor compared with that prepared with the spin-coating method. In summary, ultrasonic spray-coating technology provides promising application prospects for large-area production of functional structural color films.

## Figures and Tables

**Figure 1 polymers-13-03768-f001:**
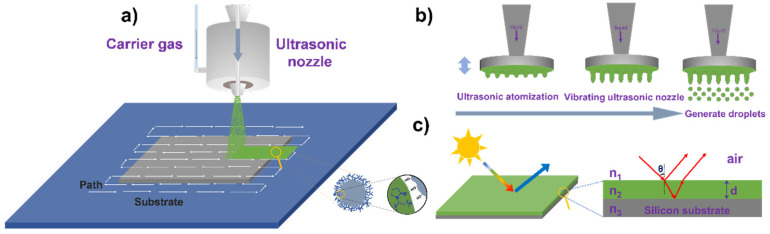
(**a**) Preparation scheme of structural color film with ultrasonic spray-coating technique, and schematic of PVP adsorption on the surface of silica particles. (**b**) The process of ultrasonic nozzle producing droplets. (**c**) Schematic of interference of the two reflected light waves from the air–film and film-silicon substrate. n_1_, n_2_, n_3_ are the refractive indices of air, SiO_2_, and silicon substrate, respectively. d is the thickness of the SiO_2_/PVP film. θ is the incident angle.

**Figure 2 polymers-13-03768-f002:**
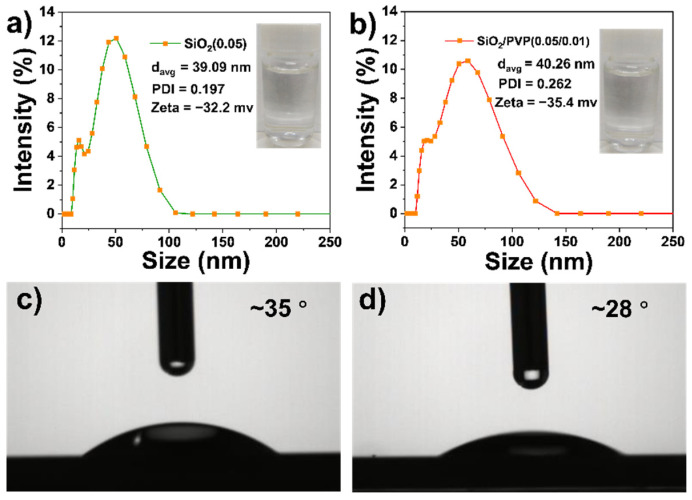
Particle size distribution of (**a**) SiO_2_ (0.05) and (**b**) SiO_2_/PVP (0.05/0.01) dispersions in water; contact angles of (**c**) SiO_2_ (0.05) and (**d**) SiO_2_/PVP (0.05/0.01) on silicon wafers.

**Figure 3 polymers-13-03768-f003:**
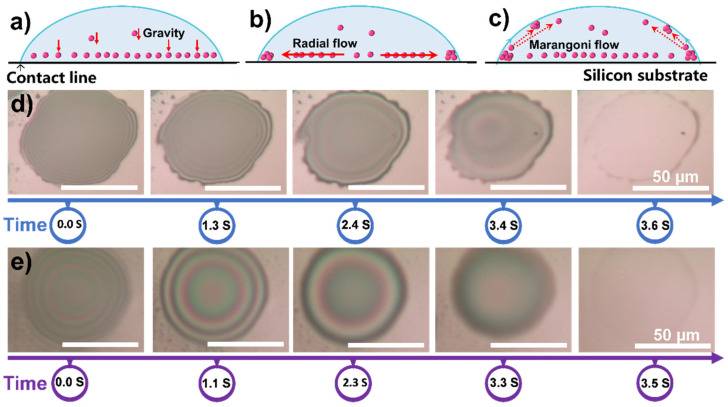
Three main flow patterns of nanoparticles in evaporated droplets of (**a**) downward flow caused by gravity, (**b**) radial flow caused by a maximum evaporation rate at the pinned wetting line and (**c**) Marangoni flow induced by surface tension differences between gas and the liquid interface. Optical microscope images of droplet drying process of (**d**) SiO_2_ (0.05) and (**e**) SiO_2_/PVP (0.05/0.01) at different times.

**Figure 4 polymers-13-03768-f004:**
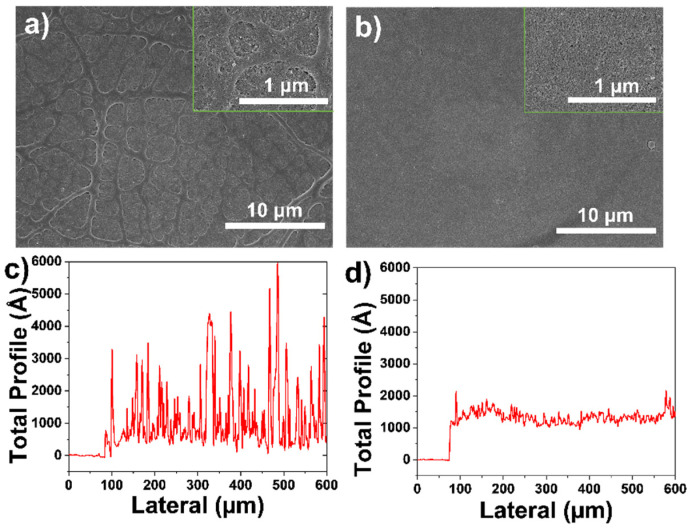
SEM images of film sprayed with (**a**) SiO_2_ (0.05) and (**b**) SiO_2_/PVP (0.05/0.01); film surface profiles of spraying solution of (**c**) SiO_2_ (0.05) and (**d**) SiO_2_/PVP (0.05/0.01).

**Figure 5 polymers-13-03768-f005:**
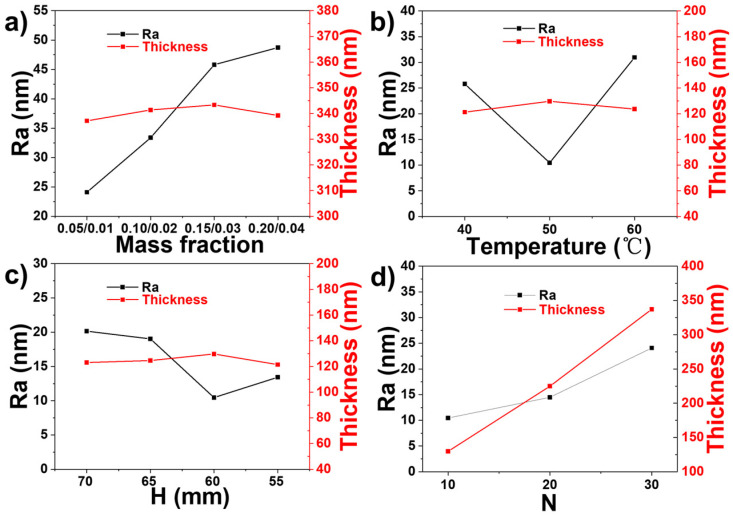
Ra and thickness of SiO_2_/PVP film with (**a**) solution concentrations of 0.05/0.01, 0.1/0.02, 0.15/0.03, and 0.2/0.04 (T = 50 °C and H = 60 mm); (**b**) temperature of = 40 °C, 50 °C, and 60 °C (concentration of 0.05/0.01 and H = 60 mm); (**c**) nozzle-to-substrate distance of 70, 65, 60, and 55 mm (concentration of 0.05/0.01 and T = 50 °C); (**d**) number of spraying passes of 10, 20, and 30 (concentration of 0.05/0.01, T = 50 °C, and H = 60 mm).

**Figure 6 polymers-13-03768-f006:**
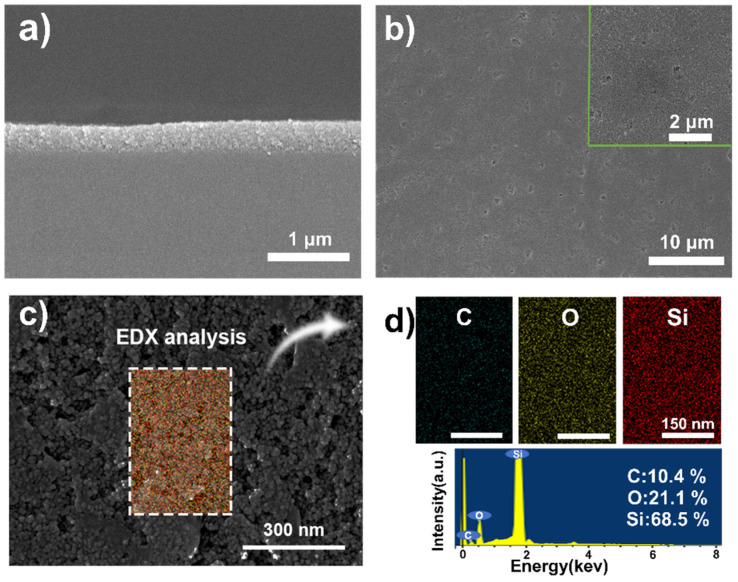
(**a**) Cross-sectional SEM image, (**b**) surface SEM image, (**c**) SEM image of high magnification, and (**d**) elemental (C, O, and Si) mappings and EDX spectrum of the nanofilm with spray passes of 30.

**Figure 7 polymers-13-03768-f007:**
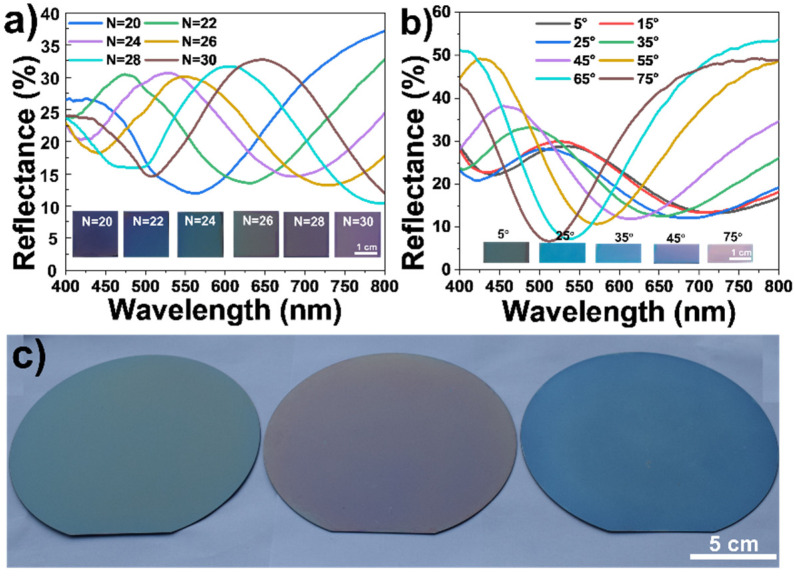
(**a**) Digital photographs and reflectance spectra. (**b**) Reflectance spectra of structural color film with spray passes of 26 at different incident angles. (**c**) Photographs of structural color films at an incident angle of 15°.

**Figure 8 polymers-13-03768-f008:**
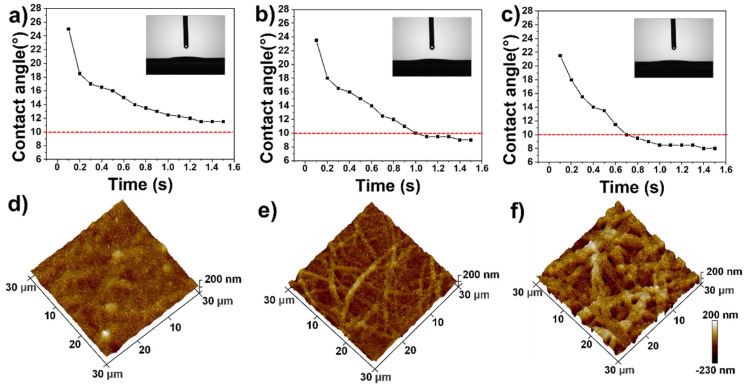
Water contact angles of structural color film with spray passes of (**a**) N = 10, (**b**) N = 20, and (**c**) N = 30 at different times; 3D AFM images of structural color film with spray passes of (**d**) N = 10, (**e**) N = 20, and (**f**) N = 30.

**Figure 9 polymers-13-03768-f009:**
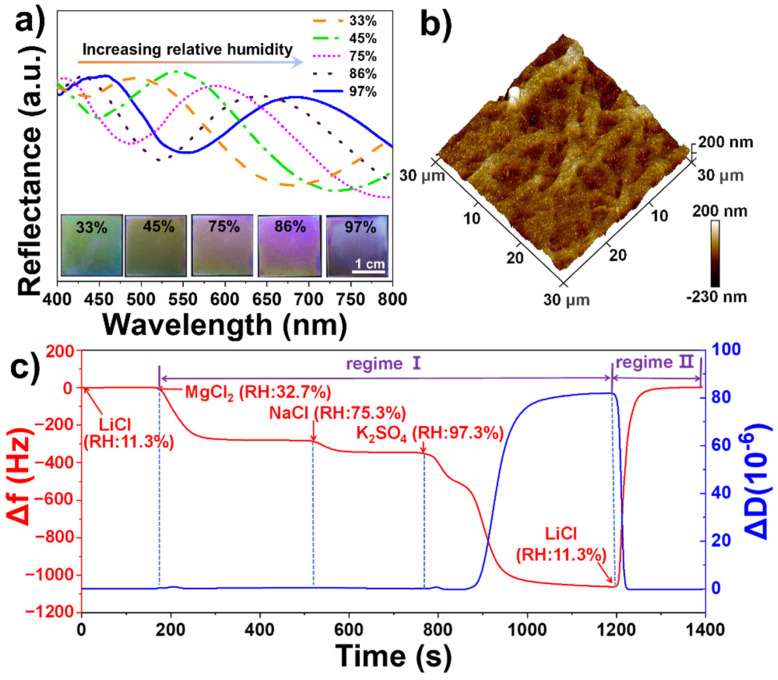
(**a**) Reflectance spectra of the structural color film with spray passes of N = 26 upon exposure to different relative humidity, (**b**) its 3D AFM image, and (**c**) variations in frequency and dissipation on a gold-coated chip when exposed to different relative humidity.

## Data Availability

All the data are available within the manuscript.

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
