# Peer review of "Large-Area Fabrication of Structurally Colored and Humidity Sensitive Composite Nanofilm via Ultrasonic Spray-Coating"

_polymers, 2021, doi:10.3390/polym13213768_

Round 1

Reviewer 1 Report

In this work fabrication of structurally colored and humidity sensitive composite nanofilm via ultrasonic spray-coating is described. Authors designed ultrasonic spraycoating technique, with non-toxic, green nano-silica and polyvinylpyrrolidone (PVP) as raw materials, to prepare structural color films on silicon wafers. It was found that the nanofilms are intelligent and show quick color change in response to humidity variation, which is attributed to their mesoporous and hydrophilic structures. Thus the developed technology provides promising application in different areas of science like nanotechnology, material science. The article looks like a short communication and may be published after minor revision.

Notes:

  1. Authors should avoid any abbreviations in the Abstract of the article.
  2. What PDI (polydispersity index) the studied dispersions of the SiO2 and SiO2/PVP have according to analyzes of the average particle size? It should be noted in the text and presented on the Figure 2 a), b).
  3. Promising application of ultrasonic spray-coating technology should be written more detailed in Conclusions. For what reason is interesting the production of functional structural color films? What areas of science are in demand in this? It should be reflected in Conclusions.

Reviewer 2 Report

In this work, "Large-area fabrication of structurally colored and humidity sensitive composite nanofilm via ultrasonic spray-coating", the authors employed ultrasonic spray-coating method to create structural color films on silicon wafers. Based on the obtained results, the authors claimed that ultrasonic spray-coating as a novel film fabrication method provides a feasible scheme for large-area structural color generation. Overall, this manuscript has a strong potential for a second review after applying the issues and addressing the shortcomings listed below:

1-The authors should polish/revise some grammatical mistakes and typos along the manuscript. I invite the authors to read their manuscript carefully and make the required changes where necessary.

2-

In the Introduction section, while discussing the recent developments in the field of structural color generation, the following works should be considered and cited to give a more general view to the possible readers of the work: [(i) Controlled self-assembly of plasmon-based photonic nanocrystals for high performance technologies, Nano Today 37, 101072 (2021); (ii) Monolithic metal dimer-on-film structure: new plasmonic properties introduced by the underlying metal, Nano Letters 20, 2087-2093 (2020)].

3-In FIGs. 2a and 2b, increase font size of the texts on axes (do the same for FIGs. 3a-3c (for the texts inside the blue region), and for FIGs. 8a-8c, FIG. 9a and 9c).

4-It seems we have two FIGs. 3. Please fix this error and make the changes accordingly. Plus, for the FIG. 3 on page 6, increase the font size of the texts on axes (do the same for FIG. 5).

5-In FIG. 7, to make FIG. 7a and FIG. 7b consistent, add the corresponding reflectance amplitude values on the y-axis of FIG. 7a (plus, add the corresponding reflectance amplitude values on the y-axis of FIG. 9a).

Reviewer 3 Report

In this paper, SiO2/PVP color thin films were prepared by ultrasonic spraying technology, and the effects of SiO2/PVP concentration, substrate temperature, distance from nozzle to substrate and spraying times on the roughness and thickness of the films were discussed. The film is sensitive to humidity changes and can quickly adjust and reversible structural color. Therefore, it is suggested that the author should revise the shortcomings of the article and put forward the following suggestions for reference.

  1. The authors should provide the reason and the optimization process of the specific content 5:1 of SiO2 and PVP ratio for research.
  2. The authors should give the repeatability test for humidity response.
  3. The response and recovery time of humidity sensing should be added and discussed.
  4. To suppress coffee-ring effect and obtain uniform film, the role of PVP is not clear and the corresponding mechanism need to be discussed and added in the section 3.1.
  5. The detailed mechanism for humidity responses of the nanofilms is missing and should be added. In addition, the humidity responses of pure SiO2 and PVP film should be researched as a comparison.

Some typos:

  1. The title of line 144 is inconsistent with the content.
  2. Line 180 should be Figure 2d instead of Figure 2e.
  3. Two Figures 3 appear in the article.
  4. The element composition in line 283 is inconsistent with the picture of the article.
  5. Check the quality of all figures and their layout problems.

Round 2

Reviewer 2 Report

In its current form, the revised manuscript is suitable for publication.

Author Response

Thank you for your recommendation!

Reviewer 3 Report

All the comments and suggestions are addressed, and I recommend the manuscript to be accepted in present form .

Author Response

Thank you for your recommendation!